# Functional, Morphological and Molecular Changes Reveal the Mechanisms Associated with Age-Related Vestibular Loss

**DOI:** 10.3390/biom13091429

**Published:** 2023-09-21

**Authors:** Vasiliki Georgia Paplou, Nick M. A. Schubert, Marcel van Tuinen, Sarath Vijayakumar, Sonja J. Pyott

**Affiliations:** 1Department of Otorhinolaryngology and Head/Neck Surgery, University Medical Center Groningen, 9713 GZ Groningen, The Netherlands; v.g.paplou@umcg.nl (V.G.P.); n.m.a.schubert@umcg.nl (N.M.A.S.); m.van.tuinen@umcg.nl (M.v.T.); 2Graduate School of Medical Sciences Research, School of Behavioural and Cognitive Neurosciences, University of Groningen, 9713 AV Groningen, The Netherlands; 3Department of Biomedical Sciences, School of Medicine, Creighton University, Omaha, NE 68178, USA; sarathvijayakumar@creighton.edu

**Keywords:** auditory, vestibular, inner ear, presbycusis or age-induced hearing loss, balance disorders, aging, inflammation, oxidative stress

## Abstract

Age-related loss of vestibular function and hearing are common disorders that arise from the loss of function of the inner ear and significantly decrease quality of life. The underlying pathophysiological mechanisms are poorly understood and difficult to investigate in humans. Therefore, our study examined young (1.5-month-old) and old (24-month-old) C57BL/6 mice, utilizing physiological, histological, and transcriptomic methods. Vestibular sensory-evoked potentials revealed that older mice had reduced wave I amplitudes and delayed wave I latencies, indicating reduced vestibular function. Immunofluorescence and image analysis revealed that older mice exhibited a significant decline in type I sensory hair cell density, particularly in hair cells connected to dimorphic vestibular afferents. An analysis of gene expression in the isolated vestibule revealed the upregulation of immune-related genes and the downregulation of genes associated with ossification and nervous system development. A comparison with the isolated cochlear sensorineural structures showed similar changes in genes related to immune response, chondrocyte differentiation, and myelin formation. These findings suggest that age-related vestibular hypofunction is linked to diminished peripheral vestibular responses, likely due to the loss of a specific subpopulation of hair cells and calyceal afferents. The upregulation of immune- and inflammation-related genes implies that inflammation contributes to these functional and structural changes. Furthermore, the comparison of gene expression between the vestibule and cochlea indicates both shared and distinct mechanisms contributing to age-related vestibular and hearing impairments. Further research is necessary to understand the mechanistic connection between inflammation and age-related balance and hearing disorders and to translate these findings into clinical treatment strategies.

## 1. Introduction

Age-related loss of vestibular function is a prevalent age-related condition that compromises the ability to perform daily tasks and diminishes the quality of life. Loss of vestibular function is characterized by unsteadiness, dizziness, and/or vertigo, as well as disturbances in gaze and postural stability [1]. The related imbalance and subsequent falls have a significant impact on both morbidity and mortality: falls account for a ten-fold increased risk of nursing home placement and are linked to serious injuries, accounting for over 50% of all accidental deaths in the elderly [2,3,4,5,6]. Age-related vestibular loss (ARVL) is also associated with an increased rate and likelihood of cognitive decline in the aged, resulting in increased odds of falling [7,8,9]. The personal, societal, and financial burden of vestibular aging is, therefore, substantial, whereas the prevention and treatment strategies are still limited. With an increase in the aging population, the prevalence of ARVL is expected to rise [10]. Therefore, an understanding of the pathophysiological changes associated with ARVL is vital to improving its prevention and management.

The vestibular system is composed of peripheral and central structures. The peripheral structures responsible for vestibular function are housed within the vestibular labyrinth or vestibule, which, alongside the cochlea responsible for auditory function, forms the inner ear. The vestibule comprises the semicircular canals that detect angular acceleration and two otolith organs—the utricle and saccule—responsible for detecting linear acceleration [11]. The neuroepithelium of the otolith organs, known as the macula, and the semicircular canals, known as the cristae ampullaris, contain type I and type II sensory hair cells and are associated supporting cells. Functioning as mechanoreceptors, these hair cells relay signals related to changes in head motion through specialized calyx and bouton afferent terminals [11]. Type I hair cells, found in the striolar/central portion of the neuroepithelium, are innervated by calyx endings derived from single afferent fibers. Type II hair cells, found more abundantly in the extrastriolar/peripheral region of the neuroepithelium, are innervated by bouton terminals from several afferents [12]. In contrast, dimorphic afferents form calyceal endings on both type I and type II hair cells and are found throughout the neuroepithelium [12]. Afferent synaptic signals from the vestibular labyrinth are conveyed via the vestibular ganglion to the vestibular nuclei in the brainstem [11,13]. The vestibular nuclei send projections into the cerebellum, spinal cord, thalamus, and nuclei of cranial nerves, contributing to adjustments of head, neck, and eye movements as well as posture, balance, and the vestibulo-ocular reflex [11,13].

Although ARVL likely involves the degeneration of both peripheral and central structures, increasing evidence in both humans and animal research models indicates that, just as for age-related hearing loss, the loss of peripheral sensorineural structures is a major contributing factor [14,15,16]. Morphological evidence from humans and animal models implicates the role of vestibular hair cell loss, particularly in the cristae, as a major cause of ARVL [17,18]. Functional measurements, specifically ocular vestibular-evoked myogenic potentials (oVEMPs) in humans, identify reduced amplitudes and increased latencies, which reflect an age-related reduction in utricular function [19,20,21]. Recent work in mice suggests the role of age-related dimorphic afferent calyceal loss in the extrastriolar region as a possible cause of age-related loss of vestibular function [22]. Possible molecular mechanisms include inflammaging, oxidative stress, and genetic factors [9]. Despite the limited number of studies investigating pathophysiological mechanisms of age-related peripheral vestibular pathology, there is still intriguing evidence of inflammaging in other disease models of aging, which can impact sensory processing [9,23]. Conditions such as cardiovascular disease, type II diabetes mellitus, and Alzheimer’s disease show evidence of chronic inflammation as well as associated ARVL [23,24,25]. The risk factors for a decline in vestibular function include smoking, hypertension, and diabetes [2]. The role of inflammaging has been more widely studied and is associated with the onset and worsening of ARHL [23,24,25].

To gain insights into the cellular and molecular mechanisms contributing to age-related vestibular loss and especially the role of inflammaging, we employed a multidimensional approach involving physiological, histological, and transcriptomic analyses to investigate the pathophysiological changes associated with aging utilizing a mouse model. Our findings reveal that ARVL is associated with diminished peripheral vestibular responses, primarily attributed to the loss of a specific subpopulation of hair cells and calyceal afferents. These findings provide valuable insights into the pathophysiological mechanisms underlying age-related vestibular and hearing impairments and lay the groundwork for future research and insights into potential diagnostic biomarkers and therapeutic interventions.

## 2. Materials and Methods

### 2.1. Animals

A total of 15 C57BL/6 mice were used in these experiments: 6 for RNA sequencing and 9 for immunofluorescent staining. Experimental protocols were approved and carried out in accordance with the relevant guidelines and regulations in place at the University Medical Center Groningen (UMCG). A total of 46 C57BL/6 mice were used as part of previously published investigations [26,27]. All mice were from the C57BL/6 background strain and used at the ages indicated.

### 2.2. Measurement of Vestibular Sensory-Evoked Potentials

Vestibular sensory-evoked potentials (VsEPs) were obtained as part of a previous investigation [27]. Young (1.5-month-old) mice ranged in age from 4 to 8 weeks old. Old (20-month-old) mice ranged in age from 81 to 95 weeks old. Mice were anesthetized with ketamine (18 mg/mL) and xylazine (2 mg/mL) injected intraperitoneally (5 to 9 μL per gram body weight). Core body temperature was maintained at 37 °C. After verification of anesthesia, the head was anchored non-invasively to a voltage-controlled mechanical shaker that provided linear acceleration pulses along the naso-occipital axis. Pulses were 2 ms in duration and delivered at a rate of 17 pulses/s. Stimulus amplitudes ranged from +6 dB to −18 dB re: 1.0 g/ms (where 1 g = 9.8 m/s^2^) adjusted in 3 dB steps. Stainless steel wire was placed subcutaneously at the nuchal crest to serve as the noninverting electrode. Needle electrodes were placed posterior to the left pinna and at the hip for inverting and ground electrodes, respectively. Responses were amplified (200,000×), filtered (300 to 3000 Hz), and digitized (100 kHz). A total of 256 primary responses were averaged for each VsEP response waveform. A broad band forward masker (from 50 to 50,000 Hz, 94 dB SPL) was presented during VsEP measurements to verify the absence of cochlear responses. VsEP threshold (wave I), amplitude (wave I or P1-N1 peak-to-peak amplitude), and latency values (onset of the stimulus to the appearance of wave I or P1) were calculated as described previously.

### 2.3. Isolation and Immunofluorescent Staining of the Vestibular Sensory Epithelia

Isolation and immunostaining of the organ of Corti and spiral ganglion neurons, microscopy, and image analysis were performed as described previously [28,29,30]. Following anesthesia via inhalation of 4% isoflurane, mice were decapitated to remove the inner ears from the temporal bones. Inner ears were then fixed in ice-cold 4% paraformaldehyde (PFA) in phosphate-buffered saline (PBS) for 1 h. After fixation, the bony labyrinths were then placed into ice-cold PBS, and the cochlear and vestibular sensory epithelia were carefully extracted. Tissue was stored in cold PBS until further processing. The isolated vestibular sensory epithelia, including the sensory epithelia of the utricle (the utricular macula) and the horizontal and anterior canals (the semicircular canal cristae), were dissected from the bony labyrinth, placed into blocking buffer (PBS with 5% normal goat serum, 4% Triton X-100, and 1% saponin) for at least 1 h, and then incubated overnight in the primary antibodies diluted 1:300 in blocking buffer. Two primary antibody combinations were used. The first combination included mouse monoclonal (IgG1) anti-CTBP2 (BD Biosciences 612044; RRID: AB_399431, San Jose, CA, USA) and rabbit polyclonal anti-GluR2/3 (Millipore AB1506; RRID: AB_90710, Burlington, MA, USA). The second combination included mouse monoclonal (IgG1) anti-Na,K-ATPase α3 (NKAα3; ThermoFisher MA3-915; RRID: AB_2274447, Waltham, MA, USA), rabbit polyclonal anti-calretinin (Millipore AB5054; RRID: AB_2068506), and mouse monoclonal (IgG2a) anti-myosin VIIa (Myo7A, Santa Cruz Biotechnology sc-74516; RRID AB_2148626, Dallas, TX, USA). Samples were rinsed 3 times for 10 min with PBS with 0.6% Triton-X 100 (PBT) and then incubated for at least 4 h in the dark with the secondary antibodies diluted 1:500 in blocking buffer. Secondary antibodies included AlexaFluor 488 goat anti-mouse (IgG1, ThermoFisher, A-21121), AlexaFluor 568 goat anti-rabbit (IgG, ThermoFisher, A-11011), and AlexaFluor 647 goat anti-mouse (IgG2a, ThermoFisher, A-21241). Samples were rinsed 3 times for 10 min with PBS with 0.6% Triton-X 100 (PBT) and 1 time in PBS before mounting and storing at 4 °C.

### 2.4. Confocal Microscopy and Quantitative Image Analysis of the Vestibular Sensory Epithelia

Confocal image z-stacks of the vestibular sensory epithelia were obtained using a Leica TCS SP8 confocal microscope. Images were taken with the 63× oil immersion objective with a resolution of 1024 × 1024 pixels, a speed of 400 Hz, and an optical zoom of 1. The z-step size (optical section thickness) was determined to be half the distance of the theoretical *z*-axis resolution (the Nyquist sampling frequency), which was approximately 0.3 µm. Quantitative image analysis was performed using Imaris v7.6. The “spots” function was used to manually mark all immunofluorescently identified hair cells within a given region of interest (ROI). The “spots” function was used to automatically mark all immunofluorescently identified pre- and postsynaptic elements within a given ROI, which were then confirmed by visual inspection.

### 2.5. Vestibule Isolation, RNA Extraction, and RNA Sequencing

Vestibule isolation, RNA extraction, and RNA sequencing were performed as described previously using only male mice [26,31]. Samples were collected at the same time of day to avoid circadian effects. Following anesthesia with inhalation of 4% isoflurane, mice were decapitated to remove their inner ears. Whole vestibules were isolated and immediately placed in TRIzol reagent and mechanically dissociated using a rotor-stator homogenizer (Tissue-Tearor). RNA isolation was performed using TRIzol and an ARCTURUS PicoPure RNA Isolation Kit. RNA isolation was followed by a DNAse step. Quality and quantity of RNA were assessed with a ThermoFisher NanoDrop and a Perkin Elmer LabChip GX. Samples with distinct RIN scores and 18S and 28S peaks were chosen for sequencing. Quality control of the sequenced samples was performed by the Genome Analysis Facility of the UMCG Genetics Department. Illumina TrueSeq RNA sample preparation kits (Illumina, Inc., Hayward, CA, USA) were used to generate sequence libraries while using the Perkin Elmer Sciclone NGS Liquid Handler (PerkinElmer, Inc., Waltham, MA, USA). Obtained cDNA fragment libraries were sequenced on an Illumina HiSeq2500 (single reads 1 × 50 bp) in pools of multiple samples. Mus musculus. GRCm38 ensembleRelease 82 reference genome was used to align the trimmed fastQ files with hisat. Sorting of aligned reads was performed using SAMtools. The gene level quantification was performed by HTSeq, and Ensembl version 82 was used as gene annotation database. FastQC was used for quality control measurements of raw sequencing data. Picard-tools calculated quality control metrics for aligned reads. RNA sequencing (RNA-seq) data were provided via a count table.

### 2.6. Differential Expression Analysis

Data were analyzed using common bioinformatic analysis pipelines in R version 4.0.5. Count tables were loaded into the R environment, and differential expression analysis was performed with the DESeq2 version 3.14 package [32]. Differential gene expression was compared between vestibules isolated from young (1.5-month-old) and old (24-month-old) C57BL/6 mice. Genes with low count numbers were excluded from analysis. Counts were r-log transformed and visualized with the ggplot2 and pheatmap packages [33,34]. Differentially expressed genes (DEGs) were defined by a false discovery rate (FDR) < 0.05. DEGs were considered upregulated with a log2foldchange (LFC) > 0 and downregulated in case of an LFC < 0. Venn diagrams were generated using an online Venn diagram tool: (http://bioinformatics.psb.ugent.be/webtools/Venn/, accessed on 20 April 2023).

### 2.7. Gene Ontology Enrichment Analysis

Gene ontology (GO) enrichment analyses were performed using the gProfiler2 web interface based on previously published protocols [35]. Two ranked gene lists, consisting of DEGs either up- or downregulated with aging, were analyzed. Statistical significance was determined using g:SCS thresholding within g:Profiler (threshold 0.05). Functional enrichment was performed on the following sources: GO cellular components, biological processes, and molecular function. Primarily enriched processes were identified in the aging vestibule. Differential changes in gene expression were then compared between the vestibule and cochlear sensorineural tissue to identify shared and unique changes in gene expression between the two substructures.

### 2.8. Statistical Analyses

All values are presented as the mean ± the standard error of the mean. Statistical analyses for transcriptomic data are described above. Statistical analyses of VsEP and imaging data were performed using one-tailed Mann–Whitney U tests since data were not always normally distributed. *p* values < 0.05 were considered statistically significant.

## 3. Results

### 3.1. Age-Related Changes in Peripheral Vestibular Function

To assess age-related changes in peripheral vestibular function, we examined the evoked potentials arising from brief head movements. The first wave (wave I) of these vestibular sensory-evoked potentials (VsEPs) is primarily generated by the activity of utricular afferent neurons [36,37]. We compared wave I thresholds, amplitudes, and latency measures between young (1.5-month-old) and old (20-month-old) mice (Figure 1). There were no significant differences in wave I thresholds between young and aged mice (Figure 1A; young: 9.9 ± 0.64 dB re: 1.0 g/ms, *n* = 14 mice; old: 8.7 ± 0.41 dB re: 1.0 g/ms, *n* = 26 mice). Even when comparing only the oldest mice (older than 20 months) to the young mice, there were no significant differences in the wave I thresholds compared to young mice. There were, however, significant differences between wave I amplitudes and latencies between young and old mice. Wave I amplitudes were significantly reduced in old compared to young mice (Figure 1B; young: 0.76 ± 0.069 μV, *n* = 14 mice; old: 0.56 ± 0.045 μV, *n* = 26 mice). Wave I latencies were significantly increased (delayed) in old compared to young mice (Figure 1C; young: 1.1 ± 0.11 ms, *n* = 14 mice; old: 1.4 ± 0.03 ms, *n* = 26 mice).

### 3.2. Age-Related Changes in the Sensorineural Structure of the Vestibular Sensory Epithelia

To determine whether alterations in peripheral vestibular function were coupled with structural changes, we used immunofluorescence to quantify changes in the numbers of sensory hair cells (Figure 2) and pre- and postsynaptic elements (Figure 3) in the striolar region of the utricular macula, the sensorineural region responsible for generating the vestibular sensory-evoked potentials [38,39,40], between young (1.5-month-old) and old (24-month-old) mice. As schematized in Figure 2A, vestibular hair cells can be divided into type I hair cells that are contacted by calyceal afferent endings and type II hair cells that are contacted by bouton afferent endings. Type I hair cells can be contacted by either calyx-only afferents or dimorphic afferents (which give rise to both calyceal and bouton afferent endings). The hair cells and afferent endings show distinct patterns of immunoreactivity.

To distinguish the various hair cell types, we took advantage of these distinct patterns of immunoreactivity (Figure 2A). Specifically, all hair cells express myosin VIIa (Myo7A; blue, Figure 2B), whereas type II hair cells preferentially express calretinin (red, Figure 2B) [41,42]. Moreover, both calyx and bouton afferent endings express Na,K-ATPase α3 (NKAα3; green, Figure 2B), whereas only calyx-only afferent terminals express calretinin (red, Figure 2B) [30,42,43,44]. Based on their combinatorial expression, we could, therefore, distinguish between type I and type II hair cells, and, moreover, between type I hair cells contacted by calyx-only afferents and those contacted by dimorphic afferents (Figure 2B). There was no qualitative difference in these patterns of immunoreactivity between young (Figure 2B upper panels) and old (Figure 2B lower panels) mice.

When examining the type I hair cell density, we found a significant reduction in old compared to young mice (Figure 2C; young: 12 ± 0.5 hair cells, *n* = 5 mice; old: 9.7 ± 0.4 hair cells, *n* = 4 mice). When further examining by the type of contacting afferent terminal, we found a significant reduction in type I hair cells contacted by dimorphic afferents in old compared to young mice (Figure 2D; young: 6.4 ± 0.8 hair cells, *n* = 5 mice; old: 4.3 ± 0.4 hair cells, *n* = 4 mice). In contrast, we found no significant difference in type I hair cells contacted by calyx-only afferents (Figure 2E; young: 4.5 ± 0.8 hair cells, *n* = 5 mice; old: 5.4 ± 0.7 hair cells, *n* = 4 mice). Furthermore, we found no significant difference in type II hair cells (Figure 2F; young: 7.7 ± 0.4 hair cells, *n* = 5 mice; old: 7.3 ± 0.6 hair cells, *n* = 4 mice).

We also used immunofluorescence to quantify changes in the density of pre- and postsynaptic elements in the striolar region of the utricular macula between young (1.5-month-old) and old mice (24-month-old; Figure 3). Presynaptic ribbons and postsynaptic glutamate receptor patches were immunostained using antibodies against CTBP2 (green, Figure 3A) and GluR2/3 (red, Figure 3A), respectively. The mean number of presynaptic ribbons in old compared to young mice was not significantly different (Figure 3B; young: 3.1 ± 0.4 ribbon cells, *n* = 5 mice; old: 2.3 ± 0.2 ribbons, *n* = 3 mice). Likewise, there was no significant difference in the density of postsynaptic glutamate receptor patches (Figure 3C; young: 3.3 ± 0.4 ribbon cells, *n* = 5 mice; old: 3.0 ± 0.2 ribbons, *n* = 3 mice). We did observe qualitative differences in the size of pre- and postsynaptic elements between young and old animals, with larger ribbons and glutamate receptor patches appearing more abundantly in young compared to old mice. These differences were not quantified.

### 3.3. Age-Related Changes in Gene Expression in the Vestibule

To investigate the mechanisms contributing to age-related changes in the function and structure of the vestibule, we used RNA sequencing to identify transcriptional differences between vestibules isolated from young (1.5-month-old) and old (24-month-old) mice. Furthermore, a comparison was made to previously examined transcriptional differences in the isolated sensorineural structures, including the organ of Corti and spiral ganglion neurons, of the cochlea [26]. Importantly, all replicates were obtained as part of a single, larger collection and sequencing workflow, thereby enabling a comparison. We first examined transcriptional differences by examining sample clustering (Figure 4A) and principal component (PC) analysis (PCA; Figure 4B). The heat map analysis showed a distinct clustering of samples by the end organ (vestibule versus cochlea) but less distinct clustering by age (young versus old; Figure 4A). PCA indicated that the most variation (PC1) was determined by transcriptional differences between end organs (vestibule versus cochlea: 64%; Figure 4B). Separation by age accounted for a much smaller fraction of the variance (PC2: 7.3%; Figure 4B).

Differential expression analysis identified 104 differentially expressed genes (DEGs) between vestibules isolated from young and old mice. DEGs were stratified into either up- or downregulated DEGs according to the log2 fold change. A total of 51 genes were found to be upregulated and 53 genes were found to be downregulated. Upregulated genes were largely associated with immunity and inflammation. Among the DEGs most upregulated with aging in the vestibule (Figure 4C) are *Ctrl*, which is a protein-coding gene related to serine-type endopeptidase and serine-type peptidase activity, predicted to be located in the extracellular space and expressed within the brain and pancreas, *Jchain*, which encodes a glycoprotein that enables IgA binding and protein homodimerization activity, contributes to immunoglobulin receptor binding, and is involved in several processes, including the defense response, and *Cxcl13*, which encodes a B lymphocyte chemoattractant and promotes the migration of B lymphocytes [45]. Notably, among the top 10 most upregulated DEGs, genes associated with inflammation and immunity included *Ccl8*, an antimicrobial gene that encodes a chemotactic factor and attracts monocytes, lymphocytes, basophils, and eosinophils, and *Ciita*, which encodes a member of the NOD-like receptor protein family and acts as a positive regulator of class II major histocompatibility complex gene transcription [45]. The complete list of DEGs is available in Appendix A.

Downregulated genes were largely associated with extracellular matrix (ECM) structural constituents, ossification, mineralization, and ECM organization. When examining the 20 most downregulated DEGs in the vestibule (Figure 4D), the top three include *Prss35*, which is a protein-coding gene located in the mitochondrion and related to serine-type endopeptidase activity and associated with pontocerebellar hypoplasia type 6, *Tnn*, which is involved in several processes, including generation of neurons and negative regulation of osteoblast differentiation, and *Col2a1*, which encodes the alpha-1 chain of type II collagen, a fibrillar collagen found in cartilage and the vitreous humor of the eye [45]. 

### 3.4. Biological Processes and Molecular Functions Associated with the Age-Related Changes in Gene Expression in the Vestibule

GO analysis based on the enrichment of gene sets within the separately analyzed up- and downregulated DEGs allowed for the identification of the biological processes and molecular functions associated with the age-related changes in gene expression in the vestibule. Notably, the enrichment analysis revealed that in the vestibule, various immunological and inflammatory processes are upregulated with aging (Figure 5). The immunological and inflammatory processes and functions included responses related to innate and adaptive immune responses, inflammatory responses, and leukocyte activation, migration, and differentiation. Notably, 25 genes were identified relating to the biological process of defense response, some of which included *Jchain*, *Cxcl13*, *Ccl8*, and *Cd74*, a protein-coding gene that enables MHC class II protein binding activity and is involved in several processes including the macrophage migration inhibitory factor signaling pathway and regulation of T cell differentiation, and *H2-Aa*, which enables peptide antigen binding activity, positive regulation of T cell regulation, and response to interferon gamma [45]. In addition, gene sets related to cell signaling and metabolism, cell development and differentiation, translation, enzyme regulation, and aging were also identified. However, these five classes contained fewer gene sets (31 gene sets combined) compared to the class containing immunological and inflammatory processes and functions (104 gene sets). Functional enrichment analyses also revealed several overrepresented cellular locations (not shown in Figure 5) that were consistent with the biological processes and molecular functions identified, including the extracellular space/region, the MHC protein complex, the complement component 1 (C1) complex, the lysosome, the IgA immunoglobulin complex, the synapse, and the ribosome. 

When examining the downregulated DEGs (Figure 6), the following categories (with total gene sets) were identified: (1) osteo- and chondrogenesis (48 gene sets), (2) cell growth and development (34 gene sets), (3) cell signaling and metabolism (26 gene sets), (4) nervous system (10 gene sets), and (5) cell migration and motility (8 gene sets). Processes related to osteo- and chondrogenesis comprised the majority of downregulated DEGs, possibly due to age-related changes identified in the bony covering of the vestibule. Analyses also revealed several cellular locations associated with downregulated DEGs (not shown in Figure 6) that were consistent with the biological processes and molecular functions identified, including ECM, cell periphery and junction, myelin, and synapse. The complete list of identified biological processes, molecular functions, and cellular locations is available in Appendix A.

### 3.5. Shared and Unique Age-Related Changes in Gene Expression between the Vestibule and Cochlea

We next investigated the overlap in transcriptional changes identified in the aging vestibule with those identified previously in the aging cochlea, and specifically the isolated sensorineural structures, including the organ of Corti and spiral ganglion neurons [26]. This previous study identified a total of 243 upregulated and 340 downregulated genes in the cochlear sensorineural structures. These DEGs were used to compare age-related changes in gene expression between the vestibule and cochlea and identify shared and unique responses to sensorineural aging.

When investigating the overlap in differential gene expression between the vestibule and cochlea, a total of 38 genes (or 37% of the DEGs identified in the vestibule) were commonly differentially expressed in both end organs (Figure 7A). A total of 26 DEGs were found to be commonly upregulated in both the vestibule and cochlea. The three DEGs included *Cxcl13*, *Cel*, and *Medag*. The protein product of *Cel* is predicted to enable several functions, including carboxylic ester hydrolase activity and glycosphingolipid binding activity. The human orthologue of this gene encodes a lipase enzyme responsible for cholesterol and lipid-soluble vitamin ester hydrolysis and absorption and associated with maturity-onset diabetes of the young (MODY), type 1 and type 2 diabetes mellitus, and the oxidation of lipoproteins to modulate atherosclerosis [45]. *Medag* encodes a cytoplasmic protein expressed in the central nervous system and retina that is predicted to be involved in positive regulation of fat cell differentiation [45]. A total of 12 DEGs were found to be commonly downregulated in both the vestibule and cochlea. The three most downregulated genes in both structures include *Col2a1*, *Snap91*, which encodes a protein that enables 1-phosphatidylinositol and clathrin binding activity and is involved in neurotransmitter secretion, and *Col11a1*, which encodes one of the two alpha chains of type XI collagen, a minor fibrillar collagen that is essential for normal embryonic skeletal development [45]. Collagen-related genes comprised a majority of the total 12 DEGs identified in both the vestibule and cochlea. The complete list of overlapping genes is available in Appendix A.

### 3.6. Biological Processes and Molecular Functions Associated with Age-Related Changes in Gene Expression Shared between the Vestibule and Cochlea

GO analysis allowed for the identification of the biological processes and molecular functions associated with shared age-related changes in gene expression between the vestibule and cochlea. Once again, the enrichment analysis revealed various immunological and inflammatory processes to be commonly upregulated with aging between the vestibule and cochlear sensorineural tissue (Figure 7B). Immunological and inflammatory processes and functions included leukocyte activation, differentiation and migration, chemotaxis, and antigen processing and presentation. Notably, 16 genes were identified comprising the biological process of defense response, including *Cxcl13*, *Cd74*, *H2-Aa*, *Rps19*, and *H2-Ab1*. *Cxcl13* and *Cd74*, which are widely recognized as inflammatory-related genes and are associated with white blood cell function and antigen presentation [46,47], were identified as being involved in the majority of immune-related processes. Additionally, gene sets associated with translation, cell migration and motility, and nervous system activation were also identified. Similar to the enriched categories identified independently in the vestibule, these three classes contained far fewer gene sets (11 gene sets combined) compared to the class containing immunological and inflammatory processes and functions (118 gene sets). 

When examining the shared downregulated DEGs (Figure 7B), gene set enrichment analysis revealed that they are largely associated with peripheral nervous system myelin formation, collagen fibril organization, myelination and axon ensheathment, chondrocyte differentiation, and cartilage condensation. The enriched categories were similar to those identified independently in the vestibule, including (1) cell growth and development (23 gene sets), (2) osteo- and chondrogenesis (17 gene sets), (3) nervous system (13 gene sets), and (4) cell signaling (4 gene sets). The complete list of identified biological processes, molecular functions, and cellular locations is available in Appendix A.

## 4. Discussion

### 4.1. Peripheral Vestibular Function Declines with Age

Functional assessment of the utricle using measurements of the vestibular sensory-evoked potentials revealed significantly reduced wave I amplitudes and delayed wave I latencies in old compared to young mice, consistent with age-related vestibular loss. Wave I thresholds were not significantly different at 24 months of age. This finding contradicts recent findings in the FVB/N mouse strain, in which vestibular thresholds were significantly elevated in equivalently old mice [22]. Strain-dependent differences in the susceptibility to age-related loss of vestibular function have been documented previously, with the C57BL/6 mouse strain we investigated showing comparatively better preservation of vestibular function despite well-recognized accelerated age-related hearing loss [48]. Our finding of normal thresholds but reduced amplitudes mirrors the phenotype of hidden hearing loss in which auditory brainstem response (ABR) wave I (cochlear) thresholds are preserved despite reduced suprathreshold wave I amplitudes [49]. Just as hidden hearing likely gives rise to hearing impairments while nevertheless evading standard audiological testing, “hidden” vestibular loss may also lead to vestibular deficits and contribute to difficulty detecting age-related vestibular loss with current clinical assessments [50].

### 4.2. Type I Vestibular Hair Cells Decline in Abundance with Age

When examining the age-related morphological changes in the utricular striolar region, the sensorineural region responsible for generating the vestibular sensory-evoked potentials [38,39,40], we observed reduced type I hair cell density, particularly in type I hair cells contacted by calyx terminals arising from dimorphic afferents. We also observed a slight but non-significant reduction in synaptic density in the utricular striolar region. Although traditional histological techniques cannot distinguish subtypes of type I hair cells by their associated afferent neurons, an abundance of studies document age-related loss of hair cells in the vestibular end organs in people and mice [9]. Our findings suggest that in old C57BL/6 mice, vestibular loss results from the age-related loss of a subset of type I hair cells and calyx terminals. The dismantlement of the calyceal junction and synaptic uncoupling between type I hair cells and calyx afferent terminals have been linked to vestibular deficits in mice in response to ototoxic exposure [28,51], and our results suggest that a similar pathology may underlie age-related vestibular loss. Additional electrophysiological evidence suggests that glutamate accumulation and spillover within the synaptic cleft between the type I hair cells and calyx terminals enhance afferent responses [52]. Thus, the age-related loss of type I hair cells and calyx terminals (despite normal synaptic densities) may specifically account for the reduced amplitudes and delayed responses of the vestibular sensory-evoked potentials observed in this study. Our results are, however, not consistent with previous examinations of equivalently aged FVB/N mice [22]. These mice showed age-related loss of synapses and calyx terminals restricted to the utricular extrastriolar region and no changes in the striolar region. Because functional assessments probe the utricular striolar region, we did not assess hair cell or synaptic density in the extrastriolar region in this study [38,39,40].

### 4.3. Age-Related Transcriptomic Changes in the Vestibule Implicate Immune and Inflammation-Related Processes

Our results demonstrate that vestibular aging is largely characterized by upregulation of immune responses and antigen processing. Upregulated genes indicate an age-related increase in inflammation, which is a hallmark of aging and multiple coexisting comorbidities linked to aging, such as cardiovascular disease, type 2 diabetes, mellitus, and Alzheimer’s disease [53]. Our results also demonstrate a significant downregulation of genes responsible for ossification, mineralization, and extracellular matrix (ECM) organization. Downregulated genes are also consistent with the well-recognized link between aging and bone loss [54,55]. With increasing age, there is a predominant shift from osteoblastogenesis to adipogenesis in the bone marrow. The lipotoxic effect of adipogenesis reduces ECM formation and mineralization, resulting in a significant reduction in bone formation. The detection of genes and processes related to osteo- and chondrogenesis are more likely linked to age-related changes in the bony epithelia of the vestibule than the vestibular neuroepithelium. The downregulation of genes associated with neuronal development, notably *Tnn*, which is involved in the generation of neurons and located in neuron projection, and *Snap91*, which is associated with neurotransmitter secretion, is consistent with age-related neurodegeneration. Finally, upregulation of genes associated specifically with aging motivates further investigation of the role of cellular senescence in age-related vestibular loss.

### 4.4. Fewer Age-Related Transcriptomic Changes in the Vestibule Compared to Cochlea

When comparing the age-related transcriptomic changes between the vestibule and cochlea, there are shared features, including some overlapping DEGs as well as associated processes, but also notable differences, including a larger number of non-overlapping DEGs and far fewer DEGs identified in the vestibule compared to the cochlea. These findings suggest that although both the vestibule and cochlea show age-related changes in transcriptomic profiles that overlap broadly with genes linked to inflammaging, the individual genes and likely the underlying mechanisms are largely unique between the two structures. Fewer age-related changes observed in the vestibule compared to the cochlea might suggest greater preservation of function and sensorineural morphology in the vestibule compared to the cochlea [9], but should be interpreted cautiously given the greater variation observed in the vestibule compared to the cochlea, which makes detecting DEGs in the vestibule more difficult. These results further suggest that age-related functional, structural, or molecular changes in one sensory system of the inner ear do not correlate with or obligate changes in the other sensory system. Additional evidence supporting an absence of association between age-related declines in vestibular and cochlear function was also recently documented in people [56].

### 4.5. Limitations

The limitations of this study are recognized. First, previous studies using equivalent functional measures found that the C57BL/6 mouse strain shows less age-related decline in vestibular function compared to cochlear function and less age-related decline in vestibular function compared to other mouse strains [57,58]. Therefore, the C57BL/6 mouse strain may give insight into the earliest stages and/or milder forms of age-related loss of vestibular function. Further investigation of other mouse strains that show greater vulnerability to age-related loss of vestibular function will be needed to better understand the progressive pathophysiology of age-related loss of vestibular function, to distinguish the shared and unique mechanisms of age-related vestibular and cochlear loss of function, and to identify the factors that modify vulnerability to age-related loss of inner ear function. Second, this study utilized transcriptomic data derived from young and old mice via bulk RNA sequencing of the entire (bone-encapsulated) vestibule, whereas functional and morphological assessments were assessed specifically in the utricle, and transcriptomic differences in the vestibule were compared to the isolated cochlear sensorineural structures without bony encapsulation. Moreover, changes in gene expression do not always correlate to changes in protein expression. Further evaluation of identified genes on a proteomic level is, therefore, recommended.

### 4.6. Clinical Implications and Future Directions

ARVL is a highly prevalent disorder, posing a tremendous individual and societal burden. The sensitivity of current tests used for detecting and diagnosing ARVL remains questionable [38]. The large inter-individual variation in disease presentation and progression makes the possibility of timely diagnosis and treatment more complex. Moreover, treatment modalities are limited, with no pharmaceutical approaches currently available to treat ARVL. Our research highlights the value of animal models to better understand the pathophysiological mechanisms and suggests areas for continued preclinical and clinical research to identify, prevent, and treat ARVL. First, our results suggest a link between ARVL and inflammation. Population-based studies could be used to examine if conditions associated with chronic inflammation are associated with an increased risk of ARVL. Identifying high-risk groups would motivate environmental or pharmaceutical interventions to help prevent the onset and minimize the burden of ARVL-associated morbidity and mortality. Second, our work identifies a number of gene products associated with ARVL that are potential pharmacological targets to halt damage progression, promote repair of the peripheral vestibular system, and ultimately result in reduced functional deficits and improved clinical management. The transcriptome-guided identification of drugs suitable for repurposing is a promising strategy for the identification of suitable drugs for repurposing [56]. 

## 5. Conclusions

The peripheral vestibular system shows functional, structural, and molecular changes associated with aging. One potential mechanism contributing to the reduced age-related vestibular function is the decreased density of type I hair cells contacted by calyx terminals arising from dimorphic afferents. Additionally, the upregulation of immune- and inflammation-related genes suggests that inflammation co-occurs with these functional and structural changes. Commonalities between vestibular and cochlear aging, namely inflammation-related processes, appear to be commonly upregulated in both the aging vestibule and cochlear sensorineural substructures. Differences between vestibular and cochlear aging were also identified. Differences indicate that distinct genes and mechanisms contribute separately to age-related vestibular and hearing impairments. Future research aimed at understanding the mechanistic connection between inflammation and age-related vestibular and hearing disorders as well as evaluating the proteomic expression of identified genes could aid in the discovery of pharmaceutical targets suitable for treatment of age-related vestibular loss.

## Figures and Tables

**Figure 1 biomolecules-13-01429-f001:**
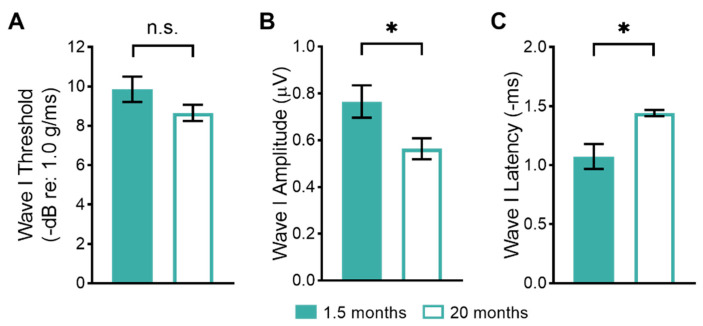
Age-related changes in peripheral vestibular function between young (1.5-month-old) and old (20-month-old) mice. (**A**). VsEP wave I thresholds are not significantly different between young and old mice. (**B**). VsEP wave I amplitudes are significantly reduced in old compared to young mice. (**C**). VsEP wave I latencies are significantly delayed in old compared to young mice. Asterisks indicate a significant difference (* denotes *p* < 0.05) as determined by the one-tailed Mann-Whitney U test. Non-significant differences are indicated by n.s.

**Figure 2 biomolecules-13-01429-f002:**
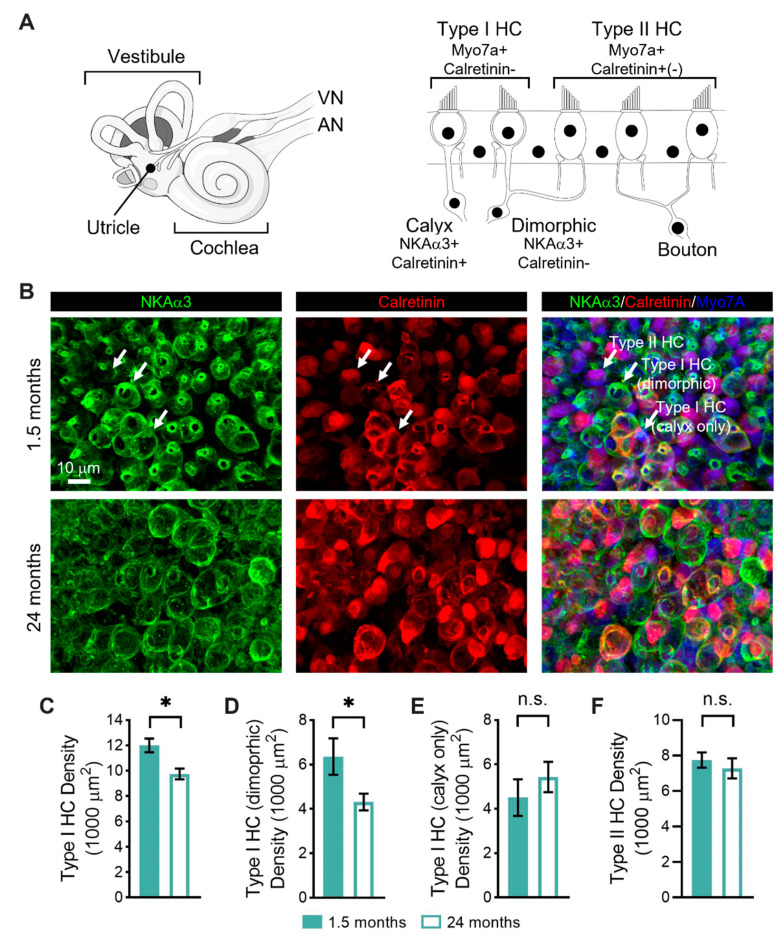
Age-related changes in the numbers and types of sensory hair cells in the vestibular sensory epithelia between young (1.5-month-old) and old (24-month-old) mice. (**A**). The inner ear contains the vestibular end organs, including the utricle, saccule, and three semicircular canals, and the cochlea, which, respectively, relay signals via the vestibular nerve (VN) and cochlear (CN) to the brain. Within the vestibular end organs, sensory hair cells can be divided into type I hair cells that are contacted by calyceal afferent endings and type II hair cells that are contacted by bouton afferent endings. Type I hair cells can be contacted by either calyx-only afferents or dimorphic afferents. The hair cells and afferent endings show distinct patterns of immunoreactivity. (**B**). Age-related changes in the numbers and types of type I and type II hair cells were investigated using immunofluorescence in the striolar region of the utricular macula. Sensory epithelia were immunostained with antibodies against Na,K-ATPase α3 (NKAα3; green) to identify all afferent endings, calretinin (red, (**B**)) to identify calyx-only afferent endings and type II hair cells, and myosin VIIa (Myo7A; blue) to identify all hair cells. Based on their combinatorial expression, type I hair cells contacted by calyx-only afferents, type I hair cells contacted by dimorphic afferents, and type II hair cells could be distinguished. No qualitative difference in these patterns of immunoreactivity were observed between young (upper panels) and old (lower panels) mice. (**C**). The density of type I hair cells was significantly reduced in old compared to young mice. (**D**). The density of type I hair cells contacted by dimorphic afferents was significantly reduced in old compared to young mice. (**E**). There was no significant difference in the density of type I hair cells contacted by calyx-only afferents. (**F**). There was no significant difference in the density of type II hair cells. Asterisks indicate a significant difference (* denotes *p* < 0.05) as determined by the one-tailed Mann-Whitney U test. Non-significant differences are indicated by n.s.

**Figure 3 biomolecules-13-01429-f003:**
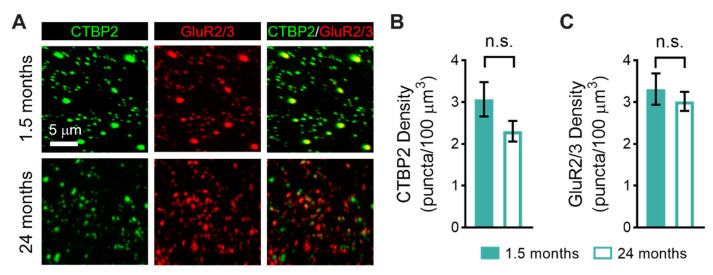
Age-related changes in the numbers of pre- and postsynaptic synaptic elements in the vestibular sensory epithelia between young (1.5-month-old) and old (24-month-old) mice. (**A**). Age-related changes in the numbers of pre-synaptic ribbons and postsynaptic glutamate receptor patches were investigated using immunofluorescence in the striolar region of the utricular macula. Sensory epithelia were immunostained with antibodies against CTBP2 (green) to identify presynaptic ribbons and GluR2/3 (red) to identify postsynaptic glutamate receptor patches (red). (**B**). There was no significant difference in the density of presynaptic ribbons (CTBP2). (**C**). There was no significant difference in the density of postsynaptic glutamate receptor patches (GluR2/3). Non-significant differences are indicated by n.s.

**Figure 4 biomolecules-13-01429-f004:**
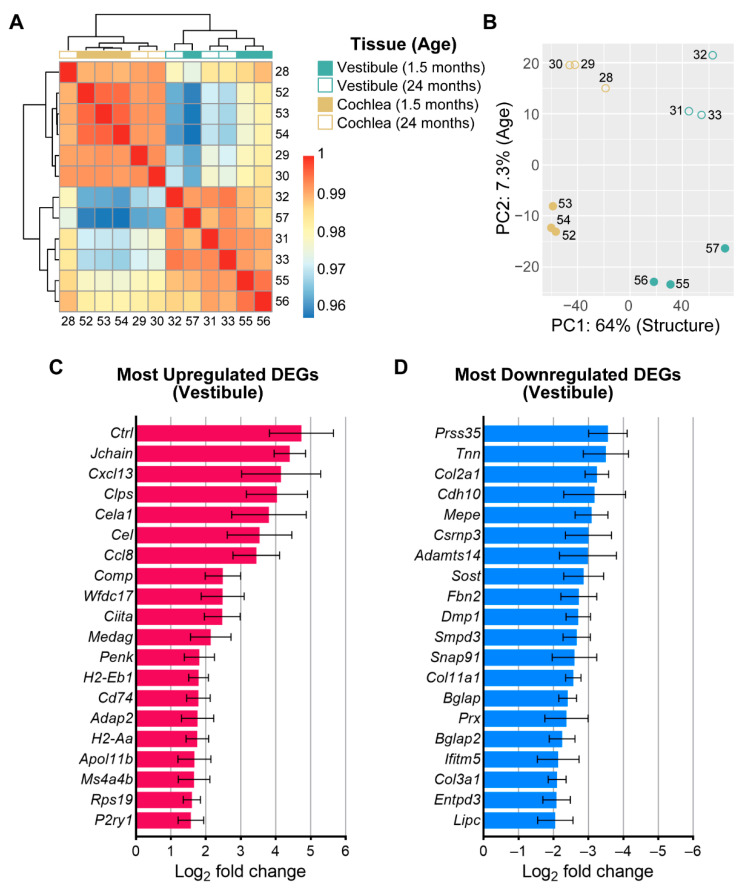
Age-related changes in gene expression in the vestibules between young (1.5-month-old) and old (24-month-old) mice. (**A**). Hierarchical clustering heatmap of Pearson correlation coefficients reveals clustering of replicates by end organ (vestibule versus cochlea) and age (young versus old). (**B**). Principal component analysis reveals that the largest variation is determined by transcriptional differences between end organs (vestibule versus cochlea; PC1: 64%) followed by age (young versus old; PC2: 7.3%). (**C**). Top 20 (of 51) genes most upregulated with age in the vestibule. (**D**). Top 20 (of 53) most downregulated genes with age in the vestibule. The complete list of DEGs is available in Appendix A.

**Figure 5 biomolecules-13-01429-f005:**
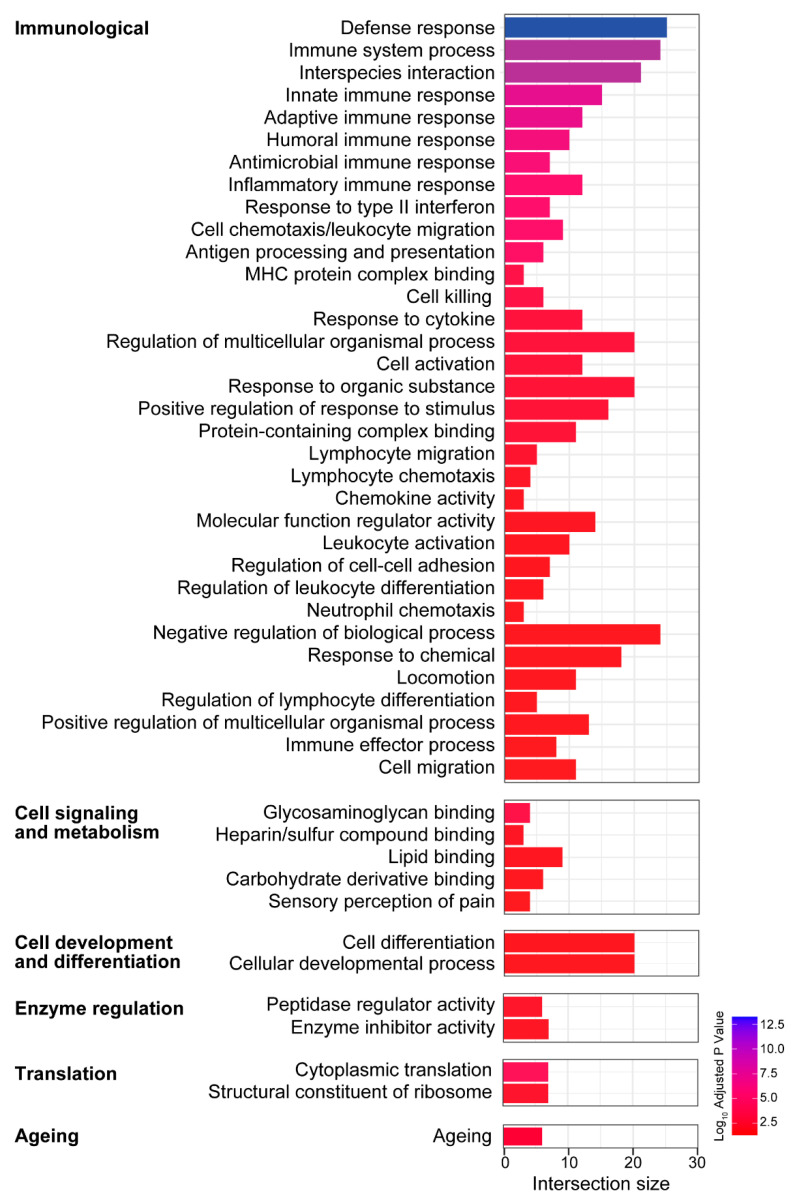
Gene ontology (GO) enrichment analysis of the genes upregulated with aging in the vestibule identifies several biological processes and molecular functions. GO terms for upregulated processes were categorized into 6 main categories: (1) immunity and inflammation, (2) cell signaling and metabolism, (3) cell development and differentiation, (4) translation, (5) enzyme regulation, and (6) aging.

**Figure 6 biomolecules-13-01429-f006:**
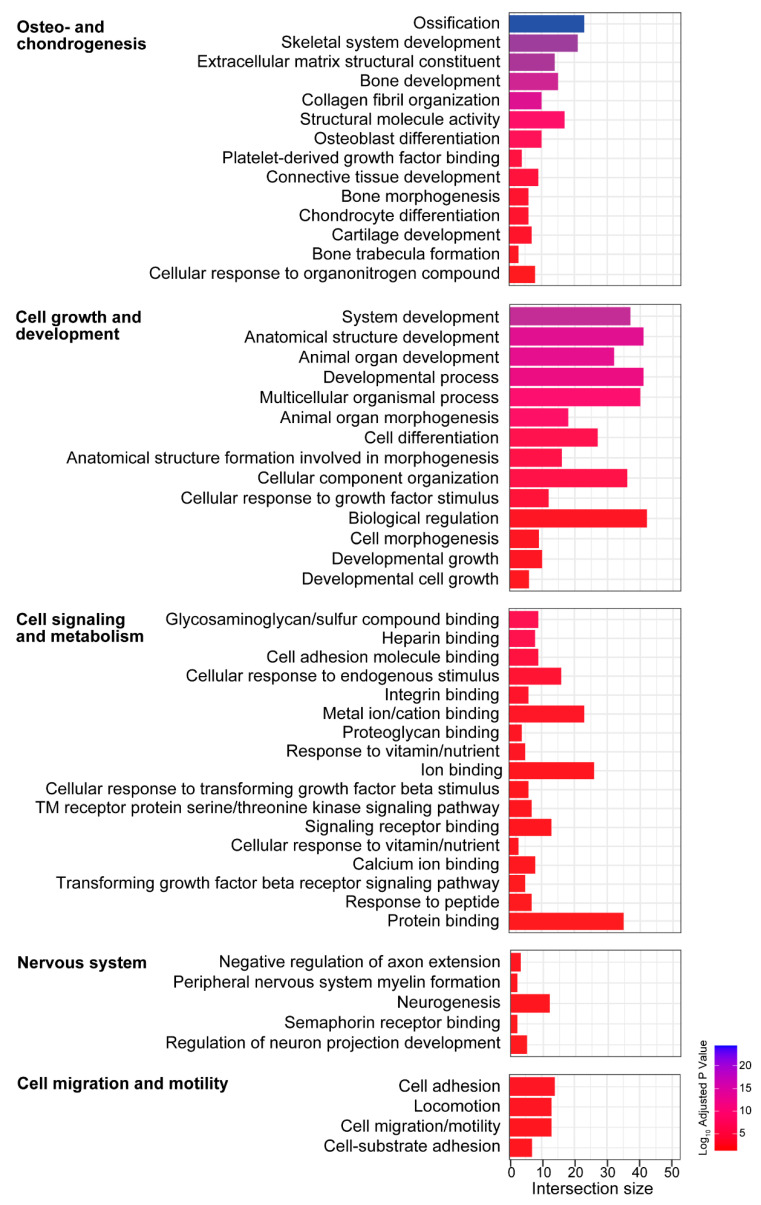
Gene ontology (GO) enrichment analysis of the genes downregulated with aging in the vestibule identifies several biological processes and molecular functions. GO terms for downregulated processes were categorized into 5 main categories: (1) osteo- and chondrogenesis, (2) cell growth and development, (3) cell signaling and metabolism, (4) nervous system, and (5) cell migration and motility.

**Figure 7 biomolecules-13-01429-f007:**
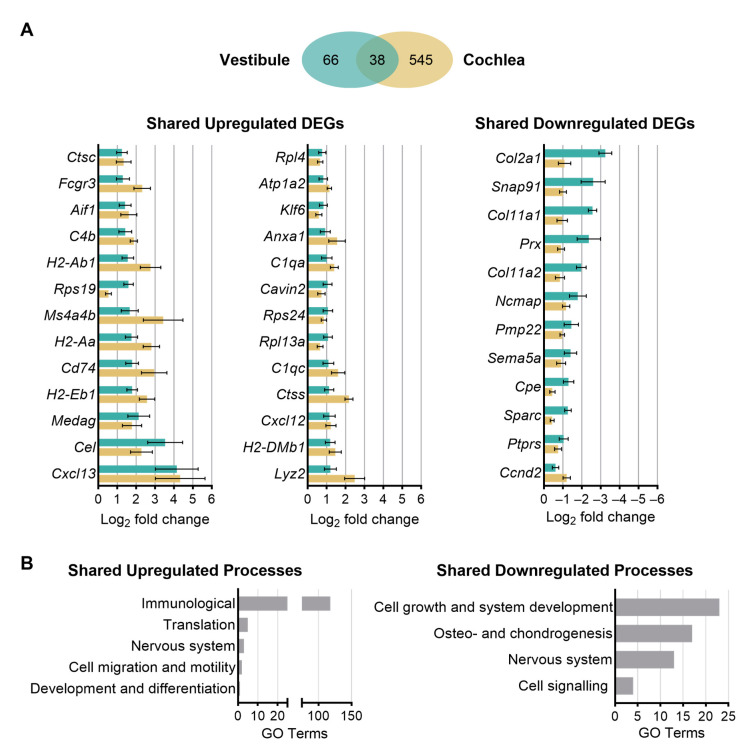
The vestibule shows distinct age-related changes in gene expression compared to cochlear sensorineural tissue. (**A**). A Venn diagram of the shared and unique genes differentially expressed with aging (DEGs) in the vestibule and cochlea. All overlapping up- and downregulated genes with aging are shown. The complete list of DEGs is available in Appendix A. (**B**). Gene ontology (GO) analysis reveals several up- and downregulated categories of molecular functions and biological processes that are commonly differentially expressed in both the vestibule and cochlear sensorineural tissue. The complete list of identified biological processes, molecular functions, and cellular locations is available in Appendix A.

## Data Availability

RNA sequencing data will be made available on the University of Maryland gene Expression Analysis Resource (gEAR) [59]. Link to be made available upon acceptance.

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
