# Peer review of "Functional, Morphological and Molecular Changes Reveal the Mechanisms Associated with Age-Related Vestibular Loss"

_biomolecules, 2023, doi:10.3390/biom13091429_

Round 1
Reviewer 1 Report
Functional, morphological, and molecular changes reveal the mechanisms associated with age-related vestibular loss by Paplou et al. is a good paper. I was interested during the reading, and liked the topic is investigated by several aspects and older (histology) and newer (gene analysis) methods, too. The schematic figures are good and help a lot. I had only one problem: I was waited more histological results – it is not a problem they had only from the utriculus but according to the introduction and the abstract I waited the other structures, too. Maybe it should be rewritten to be clearer.
Major comments:
1) Section 2.2. The description of the vestibular sensory evoked potential is very loosy. Please describe it in more detailed – what are the vestibular sensory stimulus and what type of instruments were used to detect the potentials. Were the mice awake or asleep?
Minor comments:
1) Section 3.1.: There is not latency delay. Latency is the synonym of delay, so the latency or the delay could be longer (I think).
Author Response
We appreciate the Reviewer’s positive assessment of our manuscript and time and effort providing feedback. Here below, we address these comments point-by-point.
General comment:
I was waited more histological results – it is not a problem they had only from the utriculus but according to the introduction and the abstract I waited the other structures, too. Maybe it should be rewritten to be clearer.
Response:
Examination of the other vestibular structures would be valuable as part of future research. We clarify the rationale for examining only the utricle only with additions to the text (section 3.2, page 6): “To determine whether alterations in peripheral vestibular function were coupled with structural changes, we used immunofluorescence to quantify changes in the numbers of sensory hair cells (Figure 2) and pre- and postsynaptic elements (Figure 3) in the striolar region of the utricular macula, the sensorineural region responsible for generating the vestibular sensory evoked potentials [38-40], between young (1.5-month-old) and old (24-month-old) mice.”
Major comment:
1) Section 2.2. The description of the vestibular sensory evoked potential is very loosy. Please describe it in more detailed – what are the vestibular sensory stimulus and what type of instruments were used to detect the potentials. Were the mice awake or asleep?
Response:
The description of the methods has been extended and now reads (section 2.2, page 3): “Vestibular sensory evoked potentials (VsEPs) were obtained as part of a previous investigation [27]. Young (1.5-month-old) mice ranged in age from 4 to 8 weeks old. Old (20-month-old) mice ranged in age from 81 to 95 weeks old. Mice were anesthetized with ketamine (18 mg/ml) and xylazine (2 mg/ml) injected intraperitoneally (5 to 9 μl per gram body weight). Core body temperature was maintained at 37 °C. After verification of anesthesia, the head was anchored non-invasively to a voltage-controlled mechanical shaker that provided linear acceleration pulses along the naso-occipital axis. Pulses were 2 ms in duration and delivered at a rate of 17 pulses/sec. Stimulus amplitudes ranged from +6 dB to −18 dB re: 1.0g/ms (where 1g = 9.8 m/s2) adjusted in 3 dB steps. Stainless steel wire was placed subcutaneously at the nuchal crest to serve as the noninverting electrode. Needle electrodes were placed posterior to the left pinna and at the hip for inverting and ground electrodes, respectively. Responses were amplified (200,000X), filtered (300 to 3000 Hz), and digitized (100 kHz). 256 primary responses were averaged for each VsEP response waveform. A broad band forward masker (50 to 50,000 Hz, 94 dB SPL) was presented during VsEP measurements to verify absence of cochlear responses. VsEP threshold (wave I), amplitude (wave I or P1-N1 peak-to-peak amplitude) and latency values (onset of the stimulus to the appearance of wave I or P1) were calculated as described previously.”
Minor comment:
1) Section 3.1.: There is not latency delay. Latency is the synonym of delay, so the latency or the delay could be longer (I think).
Response:
This text has been corrected and now reads (section 3.1, page 5): “Wave I latencies were significantly increased (delayed) in old compared to young mice….”
Reviewer 2 Report
Dear authors,
I really enjoyed reading your manuscript and I find it very well organised and a solid presentation of the results, setting the basis for further investigation of the cellular features and molecular causes of ARVL.
I do not have any major comment and I would suggest your publication for publication in the present form.
Best regards.
Author Response
We appreciate the Reviewer’s positive assessment of our manuscript. No additional revisions were required.
Reviewer 3 Report
This paper gives an important contribution in understanding the different mechanisms involved in the vestibular function decline related to ageing processes. A group of young mice were compared to a group of old mice using electrophysiological, histopathological and transcriptomic methods. I found the paper clear, well written and interesting for the community of scientist involved in the vestibular and hearing function. I reccommend to accept the paper as it is in the ms version that I examined
Author Response

(The authors gave the same response as above.)
